# Reading mixtures of uniform sequence-defined macromolecules to increase data storage capacity

Maximiliane Frölich[1], Dennis Hofheinz[2] & Michael A. R. Meier [1,3✉]

In recent years, the field of molecular data storage has emerged from a niche to a vibrant research topic. Herein, we describe a simultaneous and automated read-out of data stored in mixtures of sequence-defined oligomers. Therefore, twelve different sequence-defined tetramers and three hexamers with different mass markers and side chains are successfully synthesised via iterative Passerini three-component reactions and subsequent deprotection steps. By programming a straightforward python script for ESI-MS/MS analysis, it is possible to automatically sequence and thus read-out the information stored in these oligomers within one second. Most importantly, we demonstrate that the use of mass-markers as starting compounds eases MS/MS data interpretation and furthermore allows the unambiguous reading of sequences of mixtures of sequence-defined oligomers. Thus, high data storage capacity considering the field of synthetic macromolecules (up to 64.5 bit in our examples) can be obtained without the need of synthesizing long sequences, but by mixing and simultaneously analysing shorter sequence-defined oligomers.

[1] Laboratory of Applied Chemistry, Institute of Organic Chemistry (IOC), Karlsruhe Institute of Technology (KIT), Straße am Forum 7, 76131 Karlsruhe, Germany. [2] Department of Computer Science, ETH Zürich, Universitätsstrasse 6, 8092 Zürich, Switzerland. [3] Laboratory of Applied Chemistry, Institute of Biological and Chemical Systems–Functional Molecular Systems (IBCS-FMS), Karlsruhe Institute of Technology (KIT), Hermann-von-Helmholtz-Platz 1, 76344 Eggenstein-Leopoldshafen, Germany. ✉email: m.a.r.meier@kit.edu

nspired by nature, sequence-defined molecules and their application in molecular data storage and cryptography have become an important topic in the field of polymer chemistry. Such macromolecules with a defined length and distinct order of repeating units are referred to as "uniform" molecules according to IUPAC[1]. In the beginning, scientists mainly focussed on the synthesis of these highly complex structures and developed various different strategies towards uniform molecules. A lot of different synthetic pathways have been developed in the meantime, whereby an increasing degree of control and precision was achieved[2–12]. Iterative approaches are commonly used to synthesize sequence-defined molecules, since the molecules are formed step-by-step, thus enabling the highest possible control over the monomer units[6,9,13–18]. Multicomponent reactions are very suitable for sequence-definition as they offer high yields and good scalability, while they often proceed without the formation of any side products. Due to their modular character, many different functionalities can be introduced to the macromolecule as side chains[6,14,19–22]. Even dual side chain definition, offering a higher degree of definition, was demonstrated[8,23]. By using multicomponent reactions, not only the side chains, but also the backbone can be defined, thus allowing for yet another mode of dual sequence-definition (side chain and backbone) and thus further increasing the structural variety[8]. The reactions can be carried out in solution or on solid phase and are easy to conduct one-pot reactions[23]. The Passerini three component reaction (P-3CR), which was first reported in 1921 by Mario Passerini[24], is well-established in the field of sequence-defined synthesis. It is also a versatile tool for the design of different types of sequences and architectures in combination with other reactions[6,16,21,23,25–28]. The nearly quantitative yields allowed for the synthesis of long sequences with a high molecular weight in a high purity.

Recently, after many synthesis approches towards sequence-defined molecules have been developed, finding applications for this new type of molecules, for example in the field of data storage, has emerged. Like sequence-defined synthesis, the idea of molecular data storage is also inspired by nature, with DNA as prototype being vital for life as it stores our genetic code[29,30]. The DNA-sequencing of the human genome in 2001 was one of the most important achievements of the twentieth century and a breakthrough in the research field[31]. Four nucleobases, arranged in precise sequence, define the "code" that stores the genetic information of all living organisms. For artificial molecular data storage, in contrast, the binary code consisting of "0" s and "1" s is most often applied. Both DNA and synthetic sequence-defined macromolecules have in common that data is stored in long sequences of the respective repeating units in a specific order. To compare different data storage systems, the number of possible permutations is an important benchmark. A sequence of eight binary digits represents $(2^8)$ = 256 permutation, i.e., one byte. Using the quaternary system of DNA, a sequence of only four nucleobases $(4^4)$ is necessary to obtain the same number of permutations, i.e., the same amount of stored data. When synthetic molecules are used for data storage, the successful read-out of the encoded data is the most difficult step[4]. Tandem mass spectrometry seems to be a promising tool in this context and has already been successfully applied several times[4,8,32–35]. Also, other analytic methods are discussed for a possible read-out. For example, Du Prez et al. reported the use of ESI-MS for a successful read-out[2]. Until now, only a limited number of sequence-defined oligomers have been applied for the application in data storage[2,3,5,36–41]. The number of selectable functional groups per repeating unit is important for molecular data storage, as it defines the achievable number of permutations and thus the storage capacity. High data storage capacities can either be achieved by synthesizing very long sequences, as demonstrated by

Lutz and co-workers, or by increasing the information density per repeating unit[3,42]. Our group and the group of Lutz reported the first examples of using molecules with multiple selectable functionalities per repeating unit for data storage[8,20,43]. Such multiply functionalised repeating units enable to store more information in shorter oligomers, thus lowering the synthetic effort as less iterative steps are required for achieving a certain data storage capacity compared to monofunctional repeating units. The group of Du Prez presented an interesting approach based on thiolactone chemistry to synthesize sequence-defined oligomers, which were used to encode an QR code into small molecules[43]. The subsequent read-out was performed automatically using a specifically written software program. However, they reported that every oligomer had to be analysed separately, in order to reduce complexity of the resulting mass spectra, and that the automated read-out of more complex samples or mixtures would certainly increase the data storage capacity, but this would be rather difficult and requires to combine different techniques for analysis[43]. In 2020, the group of Rubenstein reported the storage of information in mixtures of small molecules[22,44]. They used the Ugi four component reaction to prepare a library of 1500 small molecules. Using a mixture, they could for instance encode a picture of Picasso with an impressively high data storage capacity of up to 0.8 million bits, with an accuracy of 97.6%[22].

In the study reported herein, the well-established stepwise iterative approach consisting of the P-3CR and a subsequent deprotection step was used[6,8,27]. Using different starter carboxylic acids and a set of twelve aldehydes for a side-chain definition, we synthesized a library of twelve different sequence-defined tetramers and three different sequence-defined hexamers. The high purity and molecular integrity of the oligomers is confirmed by mass spectrometry, nuclear magnetic resonance (NMR) as well as size exclusion chromatography (SEC), whereby SEC is able to detected impurities as low as 2% with our setup[45]. As demonstrated before by our group, characteristic fragmentation patterns are observed upon read-out by tandem mass spectrometry[8]. Based on these findings, we further enhanced the approach by incorporating a software program for an automated read-out. Using this software, the sequence order of the different tetramers and hexamers was successfully decoded. The read-out is performed in a straightforward manner within just one second by simply copying the ESI-MS/MS data into the program. The software thus represents a powerful tool, drastically facilitating the read-out. Moreover, we reported the automated read-out of a mixture of three different sequence-defined hexamers in one experiment, significantly increasing the data storage capacity compared to one single molecule. Hence, we demonstrated that even highly complex samples, like the mentioned mixture of three different sequence-defined hexamers, can be analysed automatically by tandem mass spectrometry.

## Results

**Concept and synthesis**. The data storage capacity of sequence-defined macromolecules directly correlates with the variation possibilities per repeat unit, i.e., the $X$ in the $X^Y$ notation of possible permutations, where $X$ is the base describing the available different repeat units (often different side-chains) and $Y$ is the degree of oligomerization. Thus, to achieve higher data storage capacity within sequence-defined macromolecules, either longer sequences need to be synthesized and subsequently read, as recently demonstrated for the longest sequences analysed so far[40], or the amount of data stored per repeat unit needs to be increased, as recently shown by our group for dual sequence-defined macromolecules[8]. Both approaches are challenging and laborious and certainly have synthetic limitations. For example,

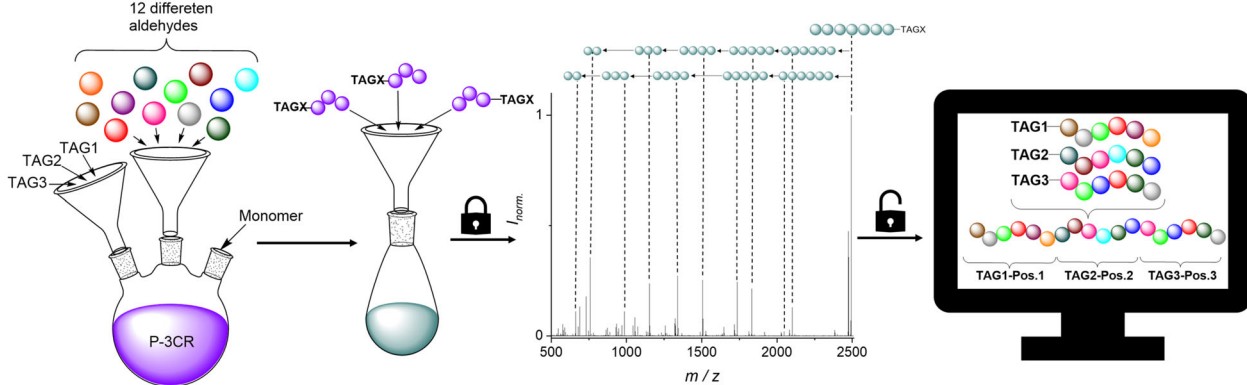

**Fig. 1 Concept of the automated read-out of a mixture of sequence-defined molecules by varying 12 different aldehydes and specifically designed mass markers (TAGs).** Iterative step synthesis with the Passerini three component reaction (P-3CR), using twelve different aldehydes and three different TAGs. The aldehydes can be introduced at any desired position of the oligomer and provide the sidechains of the macromolecule and thus differentiate each repeating unit. Subsequently, the individual sequences of an oligomer mixture can be analyzed via ESI-MS and ESI-MS/MS, followed by fully automated read-out with the computer program with a clearly defined position of the TAGs.

even if high yields in each iterative cycle were demonstrated for the approaches discussed in the introduction, the practically accessible degree of polymerization (DP) will suffer from low yields (considering that the overall yield of a 20-mer would be 12% if each iterative cycle had 90% yield). This is however not necessary, at least for data storage applications, if the sequence is established by other means (Fig. 1).

Here, molecular tags suitable for an unambiguous identification and distinction between different oligomers by high resolution MS were used. These molecular tags are used to construct a nominally longer sequence (i.e., **TAG1** defines position 1 in a virtually higher DP oligomer, and so on), based on the notion that the data storage capacity of three different hexamers is the same as that of an 18-mer, however without the associated strenuous synthesis. Thus, to achieve our goal of a simultaneous analysis of mixtures of sequence-defined oligomers, the position of the TAGs must be clearly defined for the read-out of the mixture. TAG1 is thus used to define position 1, TAG 2 defines position 2, and the same for TAG3. Specifically, designed mass markers (TAGs), herein carboxylic acids for use in a first P-3CR, were synthesized (see Supplementary Methods for detailed information). Two perfluorinated acids (**TAG1** and **TAG2**) were synthesized in a one-step synthesis as described in Supplementary Methods, while **TAG3**, a mono chlorinated acid was commercially available. For the oligomer synthesis, a stepwise iterative approach based on the Passerini three-component reaction (P-3CR) and a subsequent hydrogenolytic deprotection were applied[6]. With this synthesis protocol, various aldehydes can be used to introduce different sidechains[6,8,27]. Thus, different repeating units can be introduced to the defined oligomers at predefined positions. This procedure is well established, and all experimental details are described in the Supplementary Information (see Supplementary Methods for detailed information). Therefore, the already established isocyanide **M1** with a benzyl ester protected acid group was synthesized in a three step synthesis (see Supplementary Methods for detailed information)[6]. Using this approach, twelve different aldehydes **14a–l** (see Supplementary Methods for detailed information), to ensure side chain variation, were carefully selected to allow the simplified read-out of the sequence by tandem mass spectrometry by avoiding those aldehydes potentially yielding identical mass fragments. The aldehydes can be introduced to any desired position of the sequence, we were able to demonstrate the application of twelve different aldehydes in the synthesis of the

oligomers. Consequently, the number of the aldehydes represents the freely selectable repeating units at each position of the sequence defined oligomers. A library of twelve tetramers **T1–T12** (combining four tetramers, with each of the three **TAGs**) was prepared in a high purity (97–98%). After each reaction step of the tetramer synthesis, the products were thoroughly characterized using proton and carbon NMR spectroscopy, SEC, mass spectrometry, and infrared (IR) spectroscopy. It is noted that a tetramer can be synthesized within in 9–21 working days. One selected example of a tetramer for each TAG is displayed in Fig. 2, also providing SEC analysis after each iterative cycle of the oligomer synthesis. Furthermore, the sidechain variation of the twelve aldehydes **14a–l** (see Supplementary Methods for detailed information) in the library of the twelve tetramers **T1–T12** are visualised by the different colors of the bullets are shown in Fig. 2. The latter clearly demonstrates the successful synthesis and high purity of the prepared oligomers in a scale of 45 mg up to 2 g. For future applications, the synthesis amounts higher than required for MS/MS analysis might be needed and was thus demonstrated here. The library of the twelve tetramers with the respective SEC traces of the products, all intermediates and full characterization is reported in the Supplementary Information (Supplementary Figs. 3–146, 148–171). The three different **TAGs** were selected to increase the molar mass difference of the respective molecules to allow distinguishing molecules in mixtures during MS-experiments. Furthermore, the selection of halogenated **TAGs** provides the molecule a characteristic isotopic pattern, unique for each **TAG**, that allows to unambiguously assign the mass of an investigated oligomer to a certain TAG. Interestingly, the group of Du Prez reported the use of halogenated TAGs to write a pin code recently[2]. There, a mono-chlorinated, a mono-brominated, and a di-brominated indole were used, in addition to a non-halogenated indole. By using ESI-MS and the specific isotopic pattern, it was possible to carry out the read-out without tandem mass analysis[2]. Herein, for example the perfluorinated **TAG2** and chlorinated **TAG3** can be easily distinguished due to the characteristic isotopic pattern of the chlorinated TAG in the high resolution ESI-MS spectrum. In fact, by increasing the molecular weight of the investigated molecules, the specific isotopic pattern of the Cl-marker cannot be resolved anymore (see Supplementary Figs. 11, 18, 29, 39, 49). However, the fragmentation via ESI-MS/MS results in smaller molecular fragments, where the characteristic pattern of the Cl can be found again, which allows to distinguish the TAGs. Additionally,

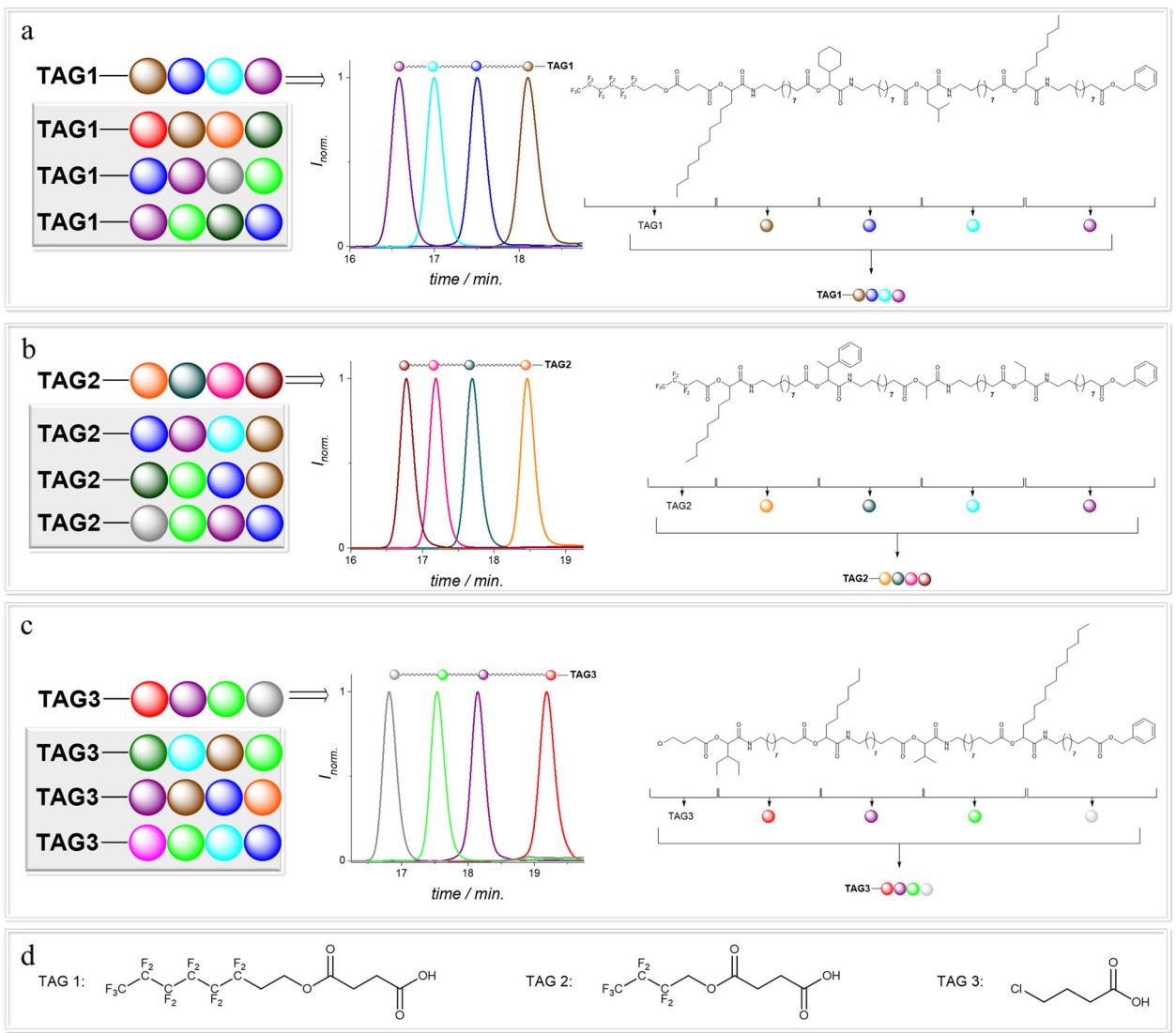

**Fig. 2 Schematic representation of the variation of the twelve different aldehydes (colored bullets) and SEC traces of three different tetramers, one for each tag. a** Chemical structure and SEC traces of a tetramer with TAG1 and the sidechain variation of the aldehydes **14a–l** (see Supplementary Methods for detailed information) for another three tetramers. **b** Chemical structure and SEC traces of a tetramer with TAG2 and the sidechain variation of the aldehydes **14a–l** (see Supplementary Methods for detailed information) for another three tetramers. **c** Chemical structure and SEC traces of a tetramer with TAG3 and the sidechain variation of the aldehydes **14a–l** (see Supplementary Methods for detailed information) for another three tetramers. **d** Chemical structures of TAG1–3.

the high mass difference between the TAGs makes them distinguishable. For the proof of concept, the mass differences between the perfluorinated **TAG1** and **TAG2** were sufficient (164 g mol$^{-1}$) to distinguish the two TAGs. This would also be possible with other commercially available acids, like stearic acid, but the use of perfluorinated acids was preferred because of the simplified workup as demonstrated previously[34].

For a mixture of three oligomers with **TAG1–3**, the mass differences and the isotopic pattern of the chlorinated TAG allowed a read-out of the oligomers. For a mixture of more than three molecules, it would be highly beneficial to introduce TAGs with another characteristic and different isotopic pattern, for instance a brominated TAG. In order to gather information on the fragmentation patterns, each of the twelve oligomers with different sidechains and tags was first analysed by tandem ESI-MS/MS. The manual analysis of the MS/MS results was important as we needed to ensure that the herein used set of aldehydes did not produce overlapping mass fragments, that would hinder the

unambiguous assignment of all peaks. Furthermore, we sought to ensure that all oligomers showed the same fragmentation patterns, independent of side groups or TAGs. The gained information was later used to write a program for a significantly accelerated, automated, and simplified read-out of the sequence-defined molecules, as described below. Thus, the program was written using the gained information about the fragmentation pattern. The storage capacity of one of the described tetramers can be calculated as follows: twelve possible side chains were taken into account, resulting in $(12)^4 = 20.736$ permutations, relating to 14.3 bit according to Eq. 2.

To further increase data storage capacity, three sequence-defined hexamers, each of which carried one of the different **TAGs**, were synthesized in order to later demonstrate the readability of a mixture of three hexamers. In Fig. 3, the chemical structure of the hexamer with **TAG1**, the SEC traces after each P-3CR and the high-resolution isotopic pattern obtained by ESI-MS compared with the calculated isotopic pattern are shown. For the

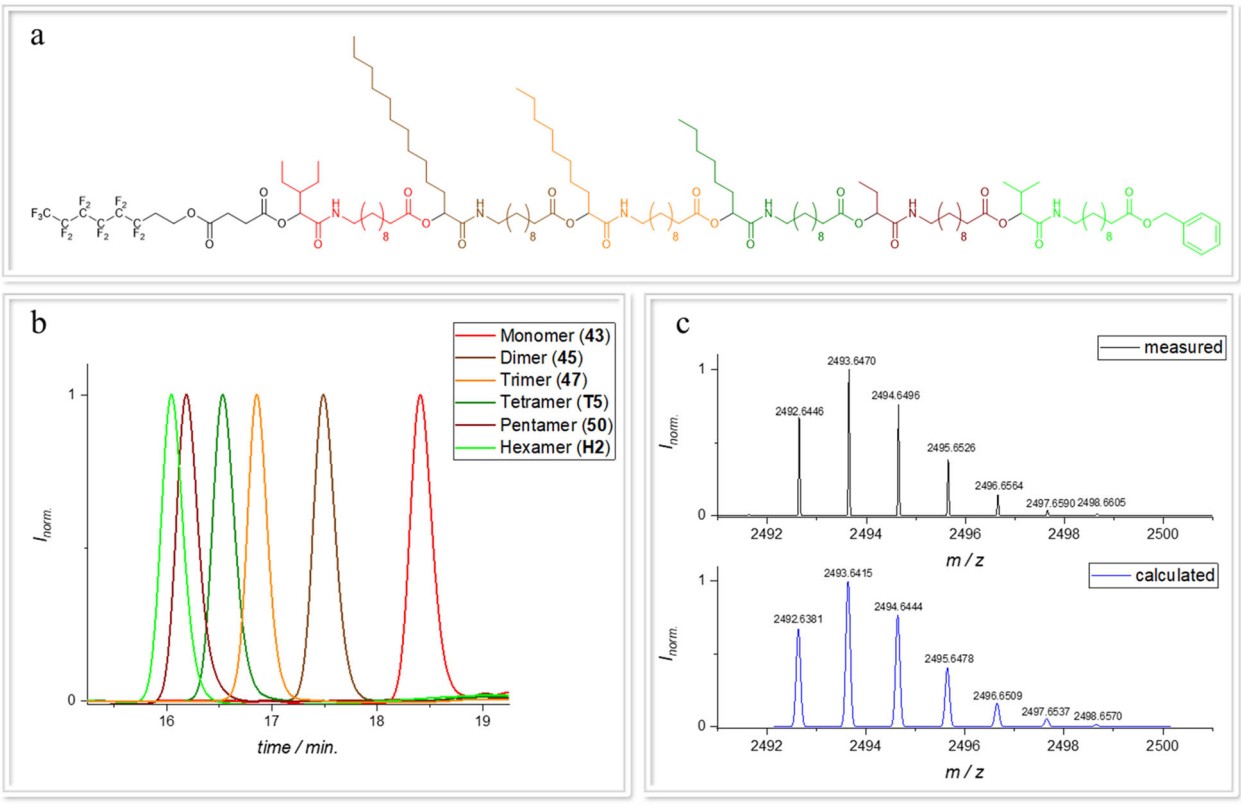

**Fig. 3 Characterization of the sequence-defined hexamer with TAG1. a** Chemical structure of the sequence-defined hexamer H2. **b** SEC traces of each P-3CR product. **c** High-resolution of the ESI-MS measurement of H2; calculated isotopic pattern (red) and measured isotopic pattern (black).

first hexamer, **TAG1** was used as starting acid. The isocyanide **M1** and 2-ethylbutyraldehyde (**14a**) (see Supplementary Methods for detailed information) were used in a small excess relative to the acid **TAG1**. The reaction was stirred at room temperature for 1 day and the Passerini product **43** (see Supplementary Methods for detailed experimental procedure) was obtained in a yield of 96.3% and high purity (98%) after a purification by column chromatography. As the oligomer grew longer, the reaction times were increased to up to six days to ensure full conversion. The first deprotection of the Passerini product **43** (see Supplementary Methods for detailed experimental procedure) was performed using ethyl acetate/THF (1:1) as solvent, and 20 wt% of the heterogeneous catalyst palladium on activated charcoal **16** (see Supplementary Methods for detailed information). The reaction was purged with hydrogen and stirred under hydrogen atmosphere overnight. The deprotected oligomers were purified via filtration over celite® to remove the heterogeneous catalyst. After eleven steps, the hexamer **H2** was obtained in an overall yield of 57.9%. All products were also analysed by NMR, SEC, mass spectrometry, and IR spectroscopy (see Supplementary Figs. 51–68).

For a successful read-out of the oligomers via tandem ESI-MS/MS, it was necessary to identify the characteristic patterns occurring during fragmentation. In a recent publication by our group, two dominant fragmentation patterns were identified: fragmentation next to the carbonyl group and next to the ester[8]. These most common dissociations are displayed in Fig. 4. Further investigations showed that the cleavage next to the ester is dominant when sodium trifluoroacetate is added during the measurement. In measurements without addition of any additive, the fragmentation next to the carbonyl is dominant. Thus, depending on the MS sample preparation, one of the two pathways can be selected and assist with the read-out and

**Fig. 4 Most common fragmentation patterns of the oligomer during fragmentation by tandem ESI-MS/MS[8]. a** Fragmentation next to the carbonyl, which is preferred in measurements without any additives. **b** The fragmentation next to the ester group is preferred when sodium trifluoroacetate is used as additive.

furthermore provide an independent read-out route for error proofing.

As an example, the hexamer **H2** is discussed here without any additive, thus the fragmentation next to the carbonyl was dominant. The tandem ESI-MS/MS fragmentation spectrum of the hexamer is displayed in Fig. 5. The mass peak of the molecule **H2** at 2492.6395 *m/z* was fragmented using a normalized collision energy (NCE) of 18. The sequence of the hexamer can be read from the left (starting at the TAG) or from the right (starting at the benzyl protection group) and the information is complementary, thus providing a further error-proof mechanism, as already discussed in our previous work[8]. Furthermore, the middle fragments without start and end blocks were observed, further confirming the structure of the molecule.

As shown in Fig. 5, the fragmentation of the oligomers follows strict rules. The reconstruction of the mass of the oligomers can

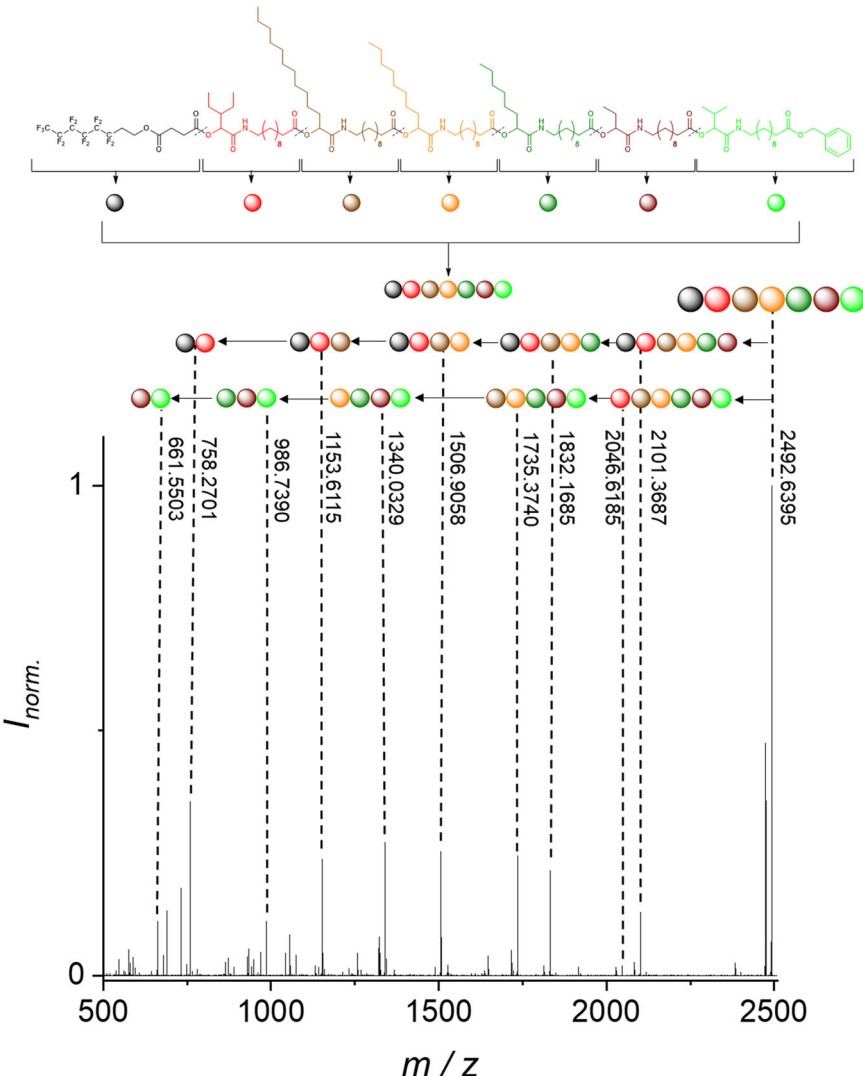

**Fig. 5 Read-out of the sequence-defined hexamer H2.** Read-out of the hexamer **H2** via tandem ESI-MS/MS with an NCE of 18. In the spectrum the read-out from both ends of the oligomer is shown, using the fragmentation next to the carbonyl.

thus be performed via the combination the masses of different fragments. Thus, the sequencing rules can be transferred to Eq. 1, providing a means for calculating the molecular mass of each fragment. More details about the equation and the exact description are shown in the Supplementary Equation 1.

$$
\begin{aligned}
&[M_{\text{Molecule}} + H]^+ \\
&= \left[ \left( M_{\text{Start}} + n^*(M_{\text{Backbone}}) + \sum_{i=1}^{i=n} M_{\text{Sidechain}}^i + M_{\text{End}} + y^*M(H) \right) + H \right]^+
\end{aligned}
\tag{1}
$$

$n$ = number of repeating units, $y = (n-1)$.

With this equation, the fundamental sequencing rules were established, allowing us to go one step further and transferring these rules to a computer program for automated read-out. The program will be discussed in detail in the following paragraph. In order to compare different data storage systems, the data storage capacity needs to be calculated. The storage capacity of one of the described hexamers is calculated as follows: twelve possible side chains were taken into account, as their suitability for this synthesis approach was demonstrated with the tetramer library and the three hexamers, as well as six repeating units, resulting in $(12)^6 = 2{,}985{,}984$ permutations. The number of permutations

can be translated into bit or byte nomenclature by using the following Eq. 2.

$$
\text{bit} = \frac{\log\left(n_{\text{permutations}}\right)}{\log(2)}.
\tag{2}
$$

Thus, the storage capacity of the described hexamers was calculated to be 21.5 bits (2.7 bytes).

**Program.** To simplify the read-out process of the oligomers and the analysis of larger molecules as well as mixtures of molecules, it was crucial to establish a computational software for MS analysis. Only a few specific fragments are necessary to re-establish the initial sequence of the oligomer, as also discussed above.

We have thus implemented a reconstruction algorithm that takes a mass spectrum as input, along with a list of masses of possible markers (**TAG1–3**), backbones (**M1**), side chains (**1a–l**) (see Supplementary Methods for detailed information), and endings (benzylgroup). The algorithm attempts to reconstruct a molecule composed out of these components, whose fragments can be found in the given mass spectrum, according to Supplementary Equation 1. Our algorithm proceeds with a directed brute force search: it enumerates all possible

combinations of side chains and compares the resulting masses with peaks from the given mass spectrum. As soon as the total mass, as well as the mass of consecutive fragments of the candidate molecule are found in the mass spectrum, the candidate is output. Our search prunes the search tree automatically, such that as soon as a considered sub fragment is not found in the mass spectrum, no molecules with that sub fragment are enumerated. Our algorithm is implemented in Python, and while we believe that significant optimizations are possible, the algorithm already takes only a few seconds on a standard laptop to correctly and uniquely analyse a mass spectrum from a molecule with six side chains. We note that Du Prez et al. also describe a related reconstruction algorithm[43]. Their algorithm uses the RDKit library (https://www.rdkit.org/) that allows to specify molecule structures in a simple and convenient manner. Like our algorithm, their algorithm also reconstructs molecules with a certain predefined structure. However, unlike our algorithm, their algorithm only matches and compares total masses of candidates and does not match against a full mass spectral information. This allows to determine molecules only up to a permutation of side chains, which in turn allows to reconstruct only significantly less information than encoded for molecules with more side chains. Moreover, Lutz et al. described a "millisecond sequencing" of binary coded polymers using a program with implemented algorithm[36,46]. This algorithm searches for the mass of the starter molecule. Afterwards, the mass of the starter plus the mass of the first backbone plus one of the possible side chains must be found. Subsequently, the next repeating unit is checked, and so on[46]. For the binary system, this easy algorithm worked well. Compared to our program, due to the use of a variety of different side chains, it was necessary to develop another algorithm. A "filter system" and some criteria, for example fragments without the mass of the TAGs, had furthermore to be implemented to our algorithm to allow for the analysis of our more complex structures.

The output of our program shows the chosen TAG and each defined sequence, with the name and the mass of the chosen aldehyde in the sidechain on the defined position. For the automatic read-out with the program, it is better to calculate three hexamers than to calculate one 18-mer, as due to the exponential increase in calculation time, an 18-mer would take too long for a calculation on a standard laptop computer.

**Read-out of three hexamers in a mixture**. As discussed above, the reading of mixtures of sequence-defined macromolecules could significantly increase that data storage capacity, while minimizing synthetic efforts and avoiding synthetic limitations. This challenge was explicitly communicated by Du Prez et al.[43]:

> "As every oligomer had to be analysed separately, a future challenge would be to combine techniques for the analysis of much more complex samples, in order to guarantee a high data density."

First of all, the read-out of each of the twelve tetramers (**T1–T12**) was performed separately to verify the successful read-out of the different TAGs (**TAG1–TAG3**) and aldehydes (**1a–l**) (see Supplementary Methods for detailed information's). After this was achieved, the analysis of more complex samples was sought. We decided to combine two different tetramers, each bearing a different TAG, to increase the data storage capacity. For the sample preparation, $0.05\,mg\,mL^{-1}$ of each tetramer were combined. Afterwards, ESI-MS of the tetramer mixture was performed. The high-resolution mass of each tetramer was analysed and compared with the calculated masses. Afterwards

each tetramer in the mixture was fragmentated via tandem ESI-MS/MS. With the combination of the two tetramers (see Supplementary Fig. 147), it was possible to increase the data storage capacity from 14.3 bit for one tetramer to 28.6 bit for the tetramer mixture, but avoiding the synthesis of an octamer, which would not be possible in the same amount of time. It is important to note here that a mixture of oligomers with the same TAGs would of course be indistinguishable. To further verify the successful read-out of a mixture of oligomers and to further increase the data storage capacity of the system, the three synthesized hexamers (**H1–H3**) were mixed. Again, the samples were prepared as a mixture with $0.05\,mg\,mL^{-1}$ of each hexamer. Then, the mixture was analysed via ESI-MS (Fig. 6a). Afterwards, the high-resolution mass of each hexamer was analysed and compared with the calculated masses. Subsequently, each hexamer in the mixture was fragmentated via tandem ESI-MS/MS. The read-out of each hexamer in the mixture is shown in Fig. 6 (for further details see Supplementary Figs. 19, 20, 67, 68, 115, 116).

For each hexamer in the mixture, the fragmentation via tandem ESI-MS/MS was performed. Afterwards, it was possible to perform the read-out of each hexamer in the mixture manually as shown in Fig. 6. To increase the data storage capacity, it was necessary to clearly define the positions of each TAGs in the mixture to generate a successful read-out, as explained above (i.e., **TAG1** for position one, and so on). Moreover, the full analysis via our above described program was successful and the output of the program shows the chosen TAG and each defined sequence, with the name and the mass of the chosen aldehyde in the sidechain on the defined position. A screenshot for the read-out of each hexamer is displayed in the Supplementary Information (Supplementary Figs. 19, 67, 115).

This full read-out of the mixture of three different hexamers allowed to increase the data storage capacity significantly from 21.5 bit for each hexamer to 64.5 bit for the mixture of the three hexamers with different TAGs. For the mixture, the permutations can be calculated as $(12)^6 *(12)^6 *(12)^6 = (12)^{18} = 26,623,333,280,885,243,904$ permutations . With Eq. 2, the number of permutations can be easily translated into 64.5 bit or 8.06 byte. As shown by the calculation, the data storage capacity of our herein used mixture of three different hexamers corresponds to the data storage capacity of one 18-mer. However, the synthesis of an 18-mer would be significantly more complex and very time consuming in comparison to the three hexamers. Furthermore, computational analysis of the 18-mer would not be feasible in this simple fashion, as it would require in the order of $2^{64}$ operations.

## Discussion

In summary, we have shown the synthesis of twelve different sequence-defined tetramers and three different hexamers with three different tags and twelve different side chains in high purity (97–98%) and yields. The oligomers were subsequently utilized for sequential read-out by tandem mass spectrometry, further demonstrating the versatility of our well-established approach. The acquired information by manual read-out of the sequences was successfully implemented in a program for automatic read-out. Using this program, the stored information of all tetramers was read automatically in a short computing time. Afterwards, the three sequence-defined hexamers were mixed and utilized for read-out. Our method, as well as the computer program, were shown to be successful and powerful tools for the automated read-out of highly complex structures and even of mixtures of different molecules. We thus developed a general method to increase the data storage capacity of sequence-defined macro-molecules by using mixtures of such compounds. The example of three hexamers provided an increase from 21.5 bit for a single hexamer up to 64.5 bit for the hexamer mixture, which

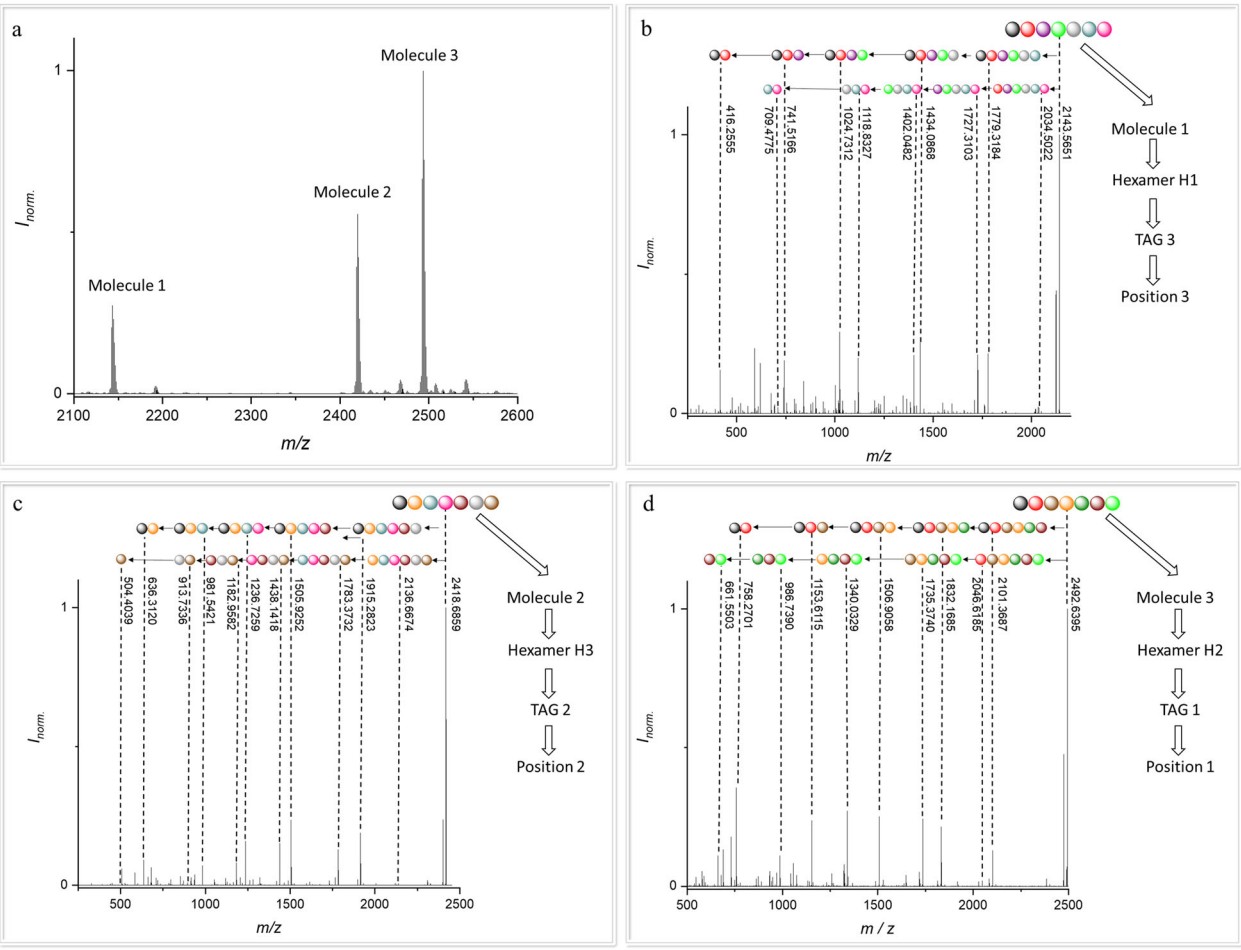

**Fig. 6 Read-out of a mixture of three hexamers, with clearly defined positions of the TAGs to increase the data storage capacity. a** ESI-MS spectrum of a mixture of three different hexamers H1-H3 that was used for subsequent tandem ESI-MS/MS fragmentation. For the fragmentation, one of the respective molecule peaks was chosen at a time. **b** fragmentation of hexamer H1. **c** fragmentation of hexamer H2. **d** fragmentation of hexamer H3.

corresponds to the data storage capacity of an 18-mer. Furthermore, the mixing of different sequence-defined oligomers (i.e., different degree of oligomerisation) as well as different numbers of oligomers (i.e., one, two or three oligomers, as demonstrated herein) for data storage allows for a straightforward adjustment of the data storage capacity without the need of further synthesis.

## Methods

**Synthesis of the perfluorinated carboxylic acid (starting block, TAG1-2).**
Detailed synthesis procedure for **TAG1** and **TAG2** as well as full characterization of all starting blocks are provided in the Supplementary Methods.

The respective perfluorinated alcohol (1.00 eq.), succinic anhydride (1.15 eq.) and DMAP (0.80 eq.) were dissolved in DCM (0.46–0.83 M). After two days of stirring at room temperature, 10 mL of DCM were added, and the solution was washed with 10% $NaHSO_4$ (ca. 15 mL). The aqueous phase was separated and extracted with DCM (2 × 60 mL). The combined organic layers were washed with water (3 × 80 mL) and dried over sodium sulfate. The solvent was evaporated under reduced pressure to obtain the perfluorinated carboxylic acid.

**Three-step synthesis of the isocyanide monomer M1.** Detailed synthesis procedure for **M1**, as well as full characterization are provided in the Supplementary Methods.

*Esterification:* 11-Aminoundecanoic acid (1.00 eq.) was suspended in THF (1 M) and benzyl alcohol (12.0 eq.) was added. The suspension was cooled in an ice bath and subsequently, thionyl chloride (3.10 eq.) was added dropwise at 0 °C. Afterwards, the solution was warmed to room temperature and stirred overnight. The yellow solution was then poured into 500 mL diethyl ether and stored in the freezer for 1 h. The product was filtered off and dried under high vacuum.

*N-formylation:* The benzyl ester ammonium chloride (1.00 eq.) was dissolved in trimethyl orthoformate (10.0 eq.) and heated to 100 °C overnight. Trimethyl

orthoformate was removed under reduced pressure and the crude product was used without further purification.

*Dehydration:* The *N*-formamide (1.00 eq.) was dissolved in DCM (0.37 M). Diisopropylamine (3.10 eq.) was added and the reaction mixture was cooled to 0 °C. Subsequently, phosphorus oxychloride (1.30 eq.) was added dropwise. Afterwards, the reaction mixture was stirred at room temperature for 2 h. The reaction was quenched by addition of sodium carbonate solution (20%, 75 mL) at 0 °C and stirring this mixture for 30 min at room temperature. 50 mL of water and 50 mL of DCM were added. The aqueous phase was separated, and the organic layer was washed with water (3 × 80 mL) and brine (80 mL). The combined organic layers were dried over sodium sulfate and the solvent was evaporated under reduced pressure. The residue was purified via column chromatography (cyclohexane/ethyl acetate) to afford the pure monomer **M1**.

**Oligomer synthesis: iterative step synthesis of P-3CR and deprotection.** The detailed synthesis procedures for each step of the oligomer syntheses as well as full characterization are provided in the Supplementary Methods.

*Passerini reaction:* The carboxylic acid **TAG1–3** (1.00 eq.) was dissolved in DCM (0.03–1.04 M). Subsequently, the aldehyde **A1–12** (1.50 eq.–3.00 eq.) and **M1** (1.50 eq.) were added. The reaction was stirred for 1–6 days at room temperature. Afterwards, and the crude product was purified via column chromatography (ethyl acetate/cyclohexane) to obtain the desired Passerini product.

*Deprotection:* The Passerini product was dissolved in a 1–1 mixture of ethyl acetate and THF. Subsequently, palladium on activated charcoal (10–20 wt%) was added. The resulting reaction mixture was purged with hydrogen gas (three balloons) and stirred overnight at room temperature under hydrogen atmosphere. The crude reaction mixture was filtered over celite® to remove the heterogeneous catalyst. The product was dried under reduced pressure to yield the free carboxylic acid.

**ESI-MS and ESI-MS/MS measurement of a mixture.** For the sample preparation, 0.05 mg mL$^{-1}$ of each hexamer **H1**–**H3** were combined. The tandem ESI-MS/MS

measurements were performed with a collision energy between NCE 15–40 and with a acquire time of ten scans. Nitrogen gas was used as collision gas.

## Data availability

All relevant data is included as Supplementary Information and is also available from the corresponding author.

## Code availability

The Pythyon script used for automated sequencing is available as Supplementary Data 1.

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

## Acknowledgements

M.F. and M.A.R.M. would like to acknowledge funding by the German Research Council (DFG) in the context of the SFB 1176 (project A3). The authors gratefully acknowledge Barbara Ridder, Rieke Schulte, Nico Zuber and Lara Faden for synthetic support and the analytical team of the Institute of Organic Chemistry for analytical support. Part of the work of D.H. was performed during his time at KIT.

## Author contributions

All authors contributed to the discussion. M.F. and M.A.R.M. conceived and designed the project. M.F. designed the experiments with input from M.A.R.M. and D.H. M.F. synthesised the oligomers and performed the ESI-MS/MS experiments. D.H. designed the program. M.F., D.H., and M.A.R.M. analysed and interpreted the data. M.F. prepared the figures. M.F. wrote the paper with feedback from all the authors.

## Funding

## Competing interests

The authors declare no competing interests.
