## [Peer Review File · Communications Chemistry]

Reviewers' comments:

Reviewer #1 (Remarks to the Author):

In this manuscript, Frölich et al. describe a data storage system based on shorter sequence-defined oligomers. In this system, 12 sequence-defined tetramers and 3 hexamers with different terminal tag and side chains were efficiently synthesized via iterative P-3CR, which offers high yields, easily functionalized and good scalability (demonstrated in ref. 6 and 16). By decoding of mixture of certain short oligomers via tandem MS/MS, i.e., mixture of 3 hexamers, high data storage capacity up to 64.5 bit can be obtained, which is much different from the conventional conception of long chain information storage. Additionally, a program or algorithm was constructed for MS analysis, which could facilitate the decoding and read-out process. This is interesting to many readers in this field, and the use of mixture of sequence defined oligomers for information storage is an interesting idea that has many possibilities for exploitation. It can be publishable after addressing the following comments.

-Generally, the conclusions reached by the authors seem plausible, however I am a bit concerned by provided theory. The authors describe that a sequence of only four nucleobases (44) has the same number of permutations as of a sequence of eight binary digits. This may be incorrect. The arrangement between each 4mer also should be considered. For example, "1111-2222-3333-4444" should be different from "2222-1111-4444-3333". Therefore, the permutations of 4 tetramer is larger than $4^4=256$. Using tags to determine the arrangement of the 4mer could be a solution. However, the authors only use TAGs to identify each oligomer in the mixture.

-In the author's system, 12 side chains can be incorporated in the polymer. Therefore, the data storage can be 124 for 4mer and 126 for 6mers. Is that means increasing the diversity of the chemical structure could provide more fragmentation pattern, thus enhance the storage capacity?

-It would be more convinced if authors could provide an example in information storage, such as encode some words in this system and read them out.

-L147 The author pointed out that the chlorinated TAG could be easily distinguished by the characteristic isotopic pattern in ESI-HRMS. However, the MS spectra of oligomers with different TAGs (such as Supplementary Figure 11 (with TAG3), Supplementary Figure 77 (with TAG1) and Supplementary Figure 107 (with TAG2)) were approximately the same. The detailed rule of the isotopic recognition in this manuscript should be specified.

-The characterization of all the 4mers and 6mers seems insufficient. It would be nice to provide the full spectrum of each tetramer and hexamer in supporting information which is much helpful to evaluate the purity and also the fidelity of the MS/MS spectra of these oligomers. Additionally, it is found that there is a lot of fragments that didn't assigned or explained in the MS/MS spectra, such as the peak next to the precursor ion in Figure 5, Figure 6 and Supplementary Figure 147.

Minor comments:

1. In Figure 2, I wonder if the color of the ball should be used according to the chemical structure, or just indicate the difference. And I think the pink ball in Figure 2c should be gray.
2. From L168 to L173, the information of products 14a, 43, 16 didn't mentioned in the main text. Therefore, readers may feel confusion. Addition of "See supporting information XX" can help readers to check the details.
3. The supporting information should be double checked. For example the hexamer in Supplementary Figure 65 should be H2 not H1. And Supplementary Figure 19 and Supplementary Figure 115 have almost the same read-out but the chemical structure of each are totally different.

Reviewer #2 (Remarks to the Author):

Overview

In this manuscript, the authors described a new framework for storing information in and reading out information from small mixtures of sequence-defined polymers. While I agree with the authors that many researchers haven't yet exploited the potentially fruitful combination of storing information in mixtures and in polymer sequences, I frankly found this paper extremely difficult to digest. First and foremost, the authors must clearly lay out what monomers they can synthesis and how many combinations of monomers are possible at each position of their polymers. I was continually confused by statements about the numbers of monomer combinations possible in this paper. I trust that the authors have done their accounting right, but I certainly could not reproduce it myself from their descriptions. I would also like to see the authors underscore the unique contributions this manuscript makes. Much of what they describe has already been done [see papers by Lutz, Rubenstein and Rosenstein, the DNA community, and the authors themselves] (perhaps except for the mixture of polymers data storage); what makes this paper new and insightful? This is not conveyed at all.

For these principle reasons and based upon the comments below, I would only recommend this paper for publications after a substantial revision focused on improving how the scientific work (which is likely high-quality) is communicated.

Substantive Comments

1. Is it fair to call 64.5 bits in the abstract "high" data storage capacity? Note that researchers have stored more than 1 MB of data in mixtures and 100s of GB of data in polymers (DNA). This wouldn't be called high data storage in comparison - and certainly not with respect to CMOS.
2. I am somewhat confused by the sentence in the Introduction "Both systems have in common that data is stored in long sequences of the respective repeating units in a specific order." I think you have to be clear about what the molecular data storage systems to which you referring are. If you are referring to polymeric molecular data storage systems, then yes, order matters. But, if you are referring to mixture-based storage systems that do not employ polymers, then order is not required.
3. I would be careful about the statement that the storage capacity of 3 hexamers is the same as a single 18-mer. The veracity of this statement depends upon whether all 3 hexamers are forced to be included in a mixture or not. If 0-3 hexamers can be included in the mixture, 3 hexamers actually provide for a larger storage capacity, as there are more possible presence/absence combinations in addition to just the hexamer sequences. Please see IEEE TRANSACTIONS ON NANOBIOSCIENCE, VOL. 19, NO. 3, JULY 2020 for a more in depth discussion of this point.
4. The article would be improved if Figure 1 more clearly indicated how many units are possible at each position to clarify exactly how many bits can be stored. This really isn't clear to me from the Figure or text. For instance, are the TAG combinations also viewed as storing information, as they should be if they are independent of the sequence? Or, are the TAG combinations completely dependent on the sequence (i.e., each TAG labels a specific, predefined sequence)? This is critical for determining the storage capacity of the polymers.
5. How are the tags mass markers if there are 4 tetramers and 3 TAGs? Shouldn't there be a unique TAG for each tetramer?
6. You also never clearly state how many possibilities the Passerini reaction yields at each position in the tetramer. Presumably 3 possibilities at each position - but then how do you only arrive at 4 unique tetramer sequences when there should be 34 possible sequences?
7. You say that there are only four (presumably unique) tetramers. Why aren't you maximizing your information storage by varying the tetramer sequences?
8. I don't understand the basis for this sentence "The manual analysis of the MS/MS results was important as we needed to ensure that the herein used set of aldehydes did not produce overlapping mass fragments that would hinder the unambiguous assignment of all peaks." What

does this have to do with analysis by-hand? Wouldn't an automated analysis yield the same result? Why can't you simply pre-check your aldehyde masses before synthesis so that the aldehydes and their adducts don't yield overlapping peaks?

9. Figure 2 seems to attempt to show the tetramer sequence variation, but confuses me even more. The key question is how many different units are possible at each monomer position? The figure seems to indicate through its colors that there are more than 3 possibilities at each position. This really needs clarification.

Where does the number 12 side chains come from in your 126 permutations? It was never clearly enunciated what the 12 units are at each position. Towards this end, I find the sentence "twelve possible side chains were taken into account, as their suitability for this synthesis approach was demonstrated with the tetramer library and the three hexamers, as well as the six repeating units..." If you have a hexamer, you have six repeating units, but if you have a tetramer, you have four. I don't understand how tetramers and hexamer can demonstrate this.

Minor Edits

1. Line 113, page 4 has a reference error

Reviewer #3 (Remarks to the Author):

The authors report on the synthesis of information-containing sequence-defined oligomers via the Passerini protocol, pioneered by this group already many years ago, in which a certain amount of data can be stored and read out by MS/MS. Whereas the synthesis and read-out as such are of no advancements, the novelty claimed in this work is that by simply mixing a few of such uniform oligomers, the data storage capacity can be significantly increased, i.e. up to 65 bit. Hence, more information can be store in shorter oligomers, thereby by-passing synthetic constraints associated to the preparation of long sequences. To enable a straightforward read-out of the discrete mixture of oligomers, specific 'mass markers' were incorporated at the start of each sequence so to allow for ESI-MS/MS characterization to be performed manually, which after successfully completion could be translated into an automated read-out script.

The main issue I have is the conceptual approach that is presented by the mixing of the sequences and the claimed increase in data storage capacity because of this. Taking into account a definition of data storage capacity as being 'the amount of data that can be stored in a storage device', I would argue that the so-called device here is the synthesised sequence-defined oligomer. Hence, simply mixing 3 tetramers, in my opinion, does not allow one to increase the data storage capacity of the macromolecules themselves. In fact, in terms of data capacity, I do not see any difference between the mixture or when one would analyse the three sequences separately. Would the stored data within one molecule not be the same in both cases? What might be claimed, however, is that the data storage density has seen an increase. Indeed, the amount of data will become more concentrated when the three solutions of oligomers are mixed in a same volume compared to the three separate solutions. Then again, a clear definition of what is interpreted as data storage density (e.g. the amount of data per mL) should be included within the current study.

Overall, the presented work is of high quality and the supporting information is technically sound and should enable to reproduce the findings. However, the claims are often massively overstated in certain places. It is therefore believed to be crucial for the authors to rework their manuscript in order to better place it with regard to the state-of-the-art in the field and also provide a more honest reflection of the advancements and claims that are made. Should this be done appropriately and the listed concerns (above and below) should have been addressed, I would highly recommend this study for publication in Communications Chemistry.

Additional comments and concerns:

1. The motivation that "the selection of halogenated TAGs provides the molecule a characteristic isotopic pattern, unique for each TAG, that allows to unambiguously assign the mass of an investigated oligomer to a certain TAG" raises an important concern. TAG3 containing a Cl is a clever choice to introduce a distinct isotopic mass pattern in HRMS [in fact, the Du Prez group has recently introduced multiple halogens in the context of molecular data storage to facilitate data read-out, yet a fair comparison/reference to this work - although cited by the authors in the introduction (ref.2) - is, albeit very relevant, missing during this discussion]. This aside, TAG1 and TAG2 should not be referred to as 'a specifically designed mass marker', given the absence of a unique finger print when using F since fluor-19 is the only abundant/stable isotope. Thus, frankly any (commercially available) starting carboxylic acid with the slightest difference in mass would be capable of providing a sufficient m/z difference. It definitely makes one question the unlogical choice for synthesising the perfluoro-alkyl acids. While indeed TAG2 can be distinguished from TAG3 based on the absence/presence of a Cl, no such differentiation is possible between TAG1 and TAG2 based on the isotopic pattern. Replacing one of these latter tags with a bromine-containing marker would significantly increase the importance of this conceptual work and should preferentially be demonstrated. In case this would not be feasible (e.g. in view of the current global situation), I believe the authors should at least address this remark within the manuscript.

2. The authors mention tandem mass spectrometry to be a promising tool to read out information contained in macromolecules. Many other techniques are also capable of extracting information, yet can be more accessible, cost-effective and render more straightforward data to interpret. It is recommended to provide the reader with at least a few other state-of-the-art techniques that have been used within the field and elaborate why tandem MS is considered most applicable in the context of this study.

3. Making oligomers longer to increase data storage capacity is claimed to have synthetic limitations because of the rapidly decreasing cumulative/overall yields that are obtained after each iterative growth cycle. This is of course correct and will be an important consideration to make, but is this anyhow problematic? Given the vast amounts of scalable protocols and the limited amount of product that is anyhow required in data storage applications, a few milligrams of a 20-mer should suffice. As such, I fail to understand why a linear synthesis of an 18-mer would be much more strenuous than making three times a linear synthesis effort to obtain three different 6-mers. It is believed important to clarify this within the manuscript.

4. On multiple occasions, the authors highlight an 'excellent', 'very high', etc. purity of the synthesised oligomers. These terms are open to interpretation and should thus be more precisely defined (i.e. expressed in %). One could argue about the use of SEC to evaluate purity, but then again the same holds for LCMS. Would a combination of both not be more appropriate to assess the presence of by-products?

5. Related to the previous concern, is there a certain requirement on the purity that needs to be reached in order to uphold the reading-out of the oligomer mixture? In other words, which levels of impurities can be tolerated during the tandem MS analysis? This is an important question to ask in order for one to justify the extensive purification efforts that have been undertaken in this study.

6. The absence of compound numbers in the manuscript (e.g. isocyanide M1, Passerini product 43) makes the discussion regarding the oligomer synthesis hard to follow. Please include compound numbers where seen fit.

7. The authors claim 'a tetramer' can be synthesised within 8 working days. This claim seems a bit purposely formulated to be misleading. Looking at the Supporting Information, T1 requires at least 14 days to synthesise (not taking into account the time needed for purification in between every growth cycle). While 8 days might be sufficient to make one certain tetramer in particular, it needs

to be specified to what compound this applies exactly. Perhaps providing a time frame that covers all prepared oligomers allows for a more honest evaluation of the synthetic efforts?

8. Upon growing the oligomers longer, much longer reaction times are needed to reach full conversion for the Passerini reaction. This is definitely a considerable drawback of this iterative growth protocol. Can these reaction times be shortened by optimising the reaction conditions (e.g. solvent, heating, catalyst)?

9. Although this does not make a difference in calculating the amount of bit that can be stored, should the max. number of possible permutations upon mixing the 3 separate hexamers not be $12^6 + 12^6 + 12^6 = 3 \cdot 12^6$, instead of $12^6 * 12^6 * 12^6 = 12^{18}$?

10. Fig. 3c: an offset instead of an overlay would make the interpretation more clear.

11. "The algorithm attempts to reconstruct a molecule composed out of these components, whose fragments can be found in the given mass spectrum, according to equation 2 "; eq. 2 allows for the calculation of the storage capacity and is not related to the read-out using the discussed algorithm. Please adapt.

12. A one-to-one comparison of the automated read-out programme is made with the one previously developed by Du Prez and co-workers. A (brief) comparison with regard to the programme by Lutz and co-workers (Macromolecules 2017, 50, 20, 8290–8296) might be a good addition to this.

Point by point:

We are thankful to all reviewers for taking time to evaluate our manuscript and for their valuable comments. We consider these comments very helpful in order to further improve the quality of our manuscript. We have taken all comments into careful consideration and have addressed the expressed concerns and suggestions point-by-point below. Changes to the manuscript are marked in green.

Reviewer 1:

In this manuscript, Frölich et al. describe a data storage system based on shorter sequence-defined oligomers. In this system, 12 sequence-defined tetramers and 3 hexamers with different terminal tag and side chains were efficiently synthesized via iterative P-3CR, which offers high yields, easily functionalized and good scalability (demonstrated in ref. 6 and 16). By decoding of mixture of certain short oligomers via tandem MS/MS, i.e., mixture of 3 hexamers, high data storage capacity up to 64.5 bit can be obtained, which is much different from the conventional conception of long chain information storage. Additionally, a program or algorithm was construct for MS analysis, which could facilitate the decoding and read-out process. This is interesting to many readers in this field, and the use of mixture of sequence defined oligomers for information storage is an interesting idea that has many possibilities for exploitation. It can be publishable after addressing the following comments.

Answer: We thank the reviewer for this summary and the overall very positive view on our manuscript.

Comment: Generally, the conclusions reached by the authors seem plausible, however I am a bit concerned by provided theory. The authors describe that a sequence of only four nucleobases (4^4) has the same number of permutations as of a sequence of eight binary digits. This may be incorrect.

Answer: Thank you for this comment. Indeed, the number of permutations is a characteristic benchmark if different data storage systems are to be compared. A sequence of eight binary digits corresponds to 1 byte (or 8 bit), which is calculated as follows: $2^8 = 256$ permutations by using the following equations.

$$(n_{\text{permutations}}) = (n_{\text{variations per repeat unit}})^{\text{Degree of Oligomerisation}}$$

$$\text{bit} = \frac{\log(n_{\text{permutations}})}{\log(2)}$$

and

$$8 \text{ bits} = 1 \text{ byte}$$

Hence, if the same number of permutations is to be achieved by DNA, only a tetramer of the four most common nucleobases is required ($4^4 = 256$ permutations) (See line 62-65 in the manuscript).

Comment: The arrangement between each 4mer also should be considered. For example, “1111-2222-3333-4444” should be different from “2222-1111-4444-3333”. Therefore, the permutations of 4 tetramer is larger than $4^4=256$. Using tags to determine the arrangement of the 4mer could be a solution. However, the authors only use TAGs to identify each oligomer in the mixture.

Answer: This is absolutely correct, but we are afraid there is a misunderstanding. To achieve the $4^4 = 256$ permutations, only one tetramer is required. The number of permutations of 4 tetramers would be $(4^4 \times 4^4 \times 4^4 \times 4^4 = 4^{16} = 4.294.967.296)$. The TAGs can indeed be used to label the respective oligomer position and thus to determine the order of different oligomers.

We added more explanation to the manuscript (line 133-134, line 330-332) as well as to the caption of Figure 1 in order to clarify this misunderstanding for future readers.

Comment: In the author’s system, 12 side chains can be incorporated in the polymer. Therefore, the data storage can be 12^4 for 4mer and 12^6 for 6mers. Is that means increasing the diversity of the chemical structure could provide more fragmentation pattern, thus enhance the storage capacity?

Answer: Indeed, by increasing the chemical diversity, the data storage capacity is increased. This can be achieved either by varying the backbone, the sidechains, or both backbone and side chain at the same time. Furthermore, the data storage capacity can be increased by elongating the sequence. Both was demonstrated by our group (*Communications Chemistry*, **2020**, 3, 63). In this publication, we compared the data storage capacity of a decamer with sidechain variation, hexamer with backbone variation and a dual sequence-defined pentamer.

Comment: It would be more convinced if authors could provide an example in information storage, such as encode some words in this system and read them out.

Answer: We appreciate this recommendation, but storing of a word or sentence in our molecules would require the definition of some kind of character encoding. In computer science, this would for instance be the well-known ASCII code. This would be possible here, but makes a direct comparison of different molecular data storage systems very difficult, as different character encoding (compare also different types of ASCII codes available) would lead to different requirement in storage capacity even for the same word, if different codes are applied. Thus, in order to be able to easily compare currently available molecular data storage systems, we prefer to stick to the current more theoretical evaluation based on permutations and stored bits of data.

Comment: L147 The author pointed out that the chlorinated TAG could be easily distinguished by the characteristic isotopic pattern in ESI-HRMS. However, the MS spectra of oligomers with different TAGs (such as Supplementary Figure 11 (with TAG3), Supplementary Figure 77 (with TAG1) and Supplementary Figure 107 (with TAG2)) were approximately the same. The detailed rule of the isotopic recognition in this manuscript should be specified.

Answer: Thank you for this comment. By increasing the molecular weight of the investigated molecules, the specific isotopic pattern of Cl is not detectable anymore. However, in smaller

molecules, like in a monomer for example, the characteristic isotope pattern is clearly observable with our MS equipment. Since we do fragmentation by tandem MS/MS, we generate smaller molecule fragments, where the characteristic pattern can be found again, which facilitates the read-out drastically as it allows to trace back the specific TAGs. In case of a tetramer, the isotope pattern of the TAG can clearly be seen in the fragmentation, as depicted below.

In order to clarify, we have added a section concerning the specific isotope pattern to the manuscript text.

In fact, by increasing the molecular weight of the investigated molecules, the specific isotopic pattern of the CI-marker cannot be resolved anymore (see supplementary Information 3.3). However, the fragmentation *via* ESI-MS/MS results in smaller molecular fragments, where the characteristic pattern of the CI can be found again, which allows to distinguish the TAGs.

Comment: The characterization of all the 4mers and 6mers seems insufficient. It would be nice to provide the full spectrum of each tetramer and hexamer in supporting information which is much helpful to evaluate the purity and also the fidelity of the MS/MS spectra of these oligomers.

Answer: In the submitted supplementary information, we provided a complete and thorough characterization after each synthetic step. Besides proton and carbon NMR, we provide HR-MS data, IR and SEC. The purity and integrity of the molecules is clearly proven by NMR, MS as well as SEC. For the sake of completeness, we have added the full MS and MS/MS spectra of the tetramers to the supplementary data.

Comment: Additionally, it is found that there is a lot of fragments that didn't assigned or explained in the MS/MS spectra, such as the peak next to the precursor ion in Figure 5, Figure 6 and Supplementary Figure 147.

Answers: This is true. In the spectra, we labeled only the most prominent fragmentation patterns that we usually find for all our sequences (explanation see lines 222-224 in the manuscript). The two main fragmentation patterns are sufficient to unambiguously read the sequence and even allows for error correction, as discussed in lines 227-228. However, also some other fragments can be observed, for instance middle-fragments. Moreover, the mentioned peak next to the precursor can be identified as [M] minus water, which can be seen in the graph below (as an example).

Minor comments:

Comment: In Figure 2, I wonder if the color of the ball should be used according to the chemical structure, or just indicate the difference. And I think the pink ball in Figure 2c should be gray.

Answer: Thank you for this comment. There was a mistake, which we corrected.

Comment: From L168 to L173, the information of products 14a, 43, 16 didn't mentioned in the main text. Therefore, readers may feel confusion. Addition of "See supporting information XX" can help readers to check the details.

Answer: Thank you, we have added a reference to the manuscript in order to clarify.

Comment: 3. The supporting information should be double checked. For example the hexamer in Supplementary Figure 65 should be H2 not H1. And Supplementary Figure 19 and Supplementary Figure 115 have almost the same read-out but the chemical structure of each are totally different.

Answer: Thank you very much for this remark. There were indeed some mistakes, which we corrected. Furthermore, we have double checked the supplementary information once more.

Reviewer 2:

Comments: In this manuscript, the authors described a new framework for storing information in and reading out information from small mixtures of sequence-defined polymers. While I agree with the authors that many researchers haven't yet exploited the potentially fruitful combination of storing information in mixtures and in polymer sequences, I frankly found this paper extremely difficult to digest. First and foremost, the authors must clearly lay out what monomers they can synthesis and how many combinations of monomers are possible at each position of their polymers.

Answer: We have carefully re-read the manuscript and also gave it to a non-specialist (a chemist not working in the field of sequence-defined macromolecules) for proof reading. This person did not find it too difficult to digest, but we nevertheless went through the manuscript sentence by sentence and tried to improve readability for authors outside the field.

I was continually confused by statements about the numbers of monomer combinations possible in this paper. I trust that the authors have done their accounting right, but I certainly could not reproduce it myself from their descriptions. I would also like to see the authors underscore the unique contributions this manuscript makes. Much of what they describe has already been done [see papers by Lutz, Rubenstein and Rosenstein, the DNA community, and the authors themselves] (perhaps except for the mixture of polymers data storage); what makes this paper new and insightful? This is not conveyed at all.

Answer: We checked the manuscript carefully and tried to better explain the terms monomer, side chain as well as permutations, and how these are related to each other. Our accounting is correct (see also comments to reviewer #1), but we hope this additional information makes reading easier.

It might seem that much has been realized yet, and this is partially true, but our system differs in many senses. For instance, Lutz typically stores data using "0" and "1" monomers, thus sticks to the binary data storage system of computers. DNA based storage technology has also achieved a lot, but is limited to 4 base pairs (compare to 12 monomers we use herein) and has chemical problems (stability, data can be copied by PCR techniques, ...). Most importantly however, as pointed out by the reviewer, we demonstrate herein the reading of mixtures, which has not been demonstrated yet for any of the above mentioned (or other) polymer data storage concepts.

For these principle reasons and based upon the comments below, I would only recommend this paper for publications after a substantial revision focused on improving how the scientific work (which is likely high-quality) is communicated.

Answer: We thank the reviewer for their generally positive response to our work. We tried our best to improve readability as suggested.

Comment: 1. Is it fair to call 64.5 bits in the abstract “high” data storage capacity? Note that researchers have stored more than 1 MB of data in mixtures and 100s of GB of data in polymers (DNA). This wouldn’t be called high data storage in comparison - and certainly not with respect to CMOS.

Answer: Thank you for this comment. We agree that 64.5 bits cannot be compared with, for example, 100s of GB of data in polymers (DNA). Our attribute “high” is however true for synthetic, i.e. non-natural, macromolecules. A comparison with commonly used IT infrastructure is on the other hand not constructive, as the field of sequence-defined macromolecules is still relatively young. Thus, we rephrased and clearly mentioned that this data storage capacity is high only in this particular field, i.e. synthetic macromolecules.

Comment: 2. I am somewhat confused by the sentence in the Introduction “Both systems have in common that data is stored in long sequences of the respective repeating units in a specific order.” I think you have to be clear about what the molecular data storage systems to which you referring are. If you are referring to polymeric molecular data storage systems, then yes, order matters. But, if you are referring to mixture-based storage systems that do not employ polymers, then order is not required.

Answer: Thank you for this comment, the mentioned sentence was indeed not self-explanatory. We changed the sentence as follows:

“Both DNA and synthetic sequence-defined macromolecules have in common that data is stored in long sequences of the respective repeating units in a specific order.”

Comment: 3. I would be careful about the statement that the storage capacity of 3 hexamers it the same as a single 18-mer. The veracity of this statement depends upon whether all 3 hexamers are forced to be included in a mixture or not. If 0-3 hexamers can be included in the mixture, 3 hexamers actually provide for a larger storage capacity, as there are more possible presence/absence combinations in addition to just the hexamer sequences. Please see IEEE TRANSACTIONS ON NANOBIOSCIENCE, VOL. 19, NO. 3, JULY 2020 for a more in depth discussion of this point.

Answers: We agree with this comment. For our statement to be valid, it is necessary that three hexamers are included in the mixture, and that they are labelled with position TAGs. We clarified this in the manuscript text (line 133-134, line 330-332).

The given reference describes a complete issue of IEEE TRANSACTIONS ON NANOBIOSCIENCE, we assume the reviewer wanted to draw our attention to manuscript DOI: 10.1109/TNB.2020.2977304, where mixtures of molecules were used to store data (see line

85-88). Thus, the general idea is somewhat related to this manuscript. As an application, a binary image was encoded showing minor errors during read-out. The latter is absolutely unproblematic for the described application, but shows the advantage of our system, i.e. 100% accurate reading with included error correction (if necessary). Depending on the targeted application, both systems have relevance and the user can decide for their requirements.

Comment: 4. The article would be improved if Figure 1 more clearly indicated how many units are possible at each position to clarify exactly how many bits can be stored. This really is not clear to me from the Figure or text.

Answers: Thank you for this comment. We modified Figure 1 accordingly. It now shows that it is possible to use 12 different aldehydes (meaning 12 possibilities for each unit). The number of permutations can then be calculated using the following formula:

$$(n_{\text{permutations}}) = (n_{\text{number of aldehydes}})^{\text{repeating units}}$$

For a tetramer, this would correspond to 12^4 permutations. Afterwards, the number of bits can be calculated from the number of permutations and Equation 2 (see manuscript).

Comment: For instance, are the TAG combinations also viewed as storing information, as they should be if they are independent of the sequence? Or, are the TAG combinations completely dependent on the sequence (i.e., each TAG labels a specific, predefined sequence)? This is critical for determining the storage capacity of the polymers.

Answers: The TAGs are independent of the sequence of the individual macromolecule. They are used as a marker to define the order of the respective sequence (i.e. the 3 hexamers) to enable the “combination” to an even longer sequence. For calculating the data storage capacity of one single oligomer, the TAGs are not considered. To improve readability, we changed the caption of Figure 1 and added an explanation of the TAGs:

The aldehydes can be introduced at any desired position of the oligomer and provide the sidechains of the macromolecule and thus differentiate each repeating unit. Subsequently, the individual sequences of an oligomer mixture can be analyzed *via* ESI-MS and ESI-MS/MS, followed by fully automated read-out with the computer program with a clearly defined position of the TAGs.

Comment: 5. How are the tags mass markers if there are 4 tetramers and 3 TAGs? Shouldn't there be a unique TAG for each tetramer?

Answer: This is an important question. The simple answer is: We have never performed a measurement of a mixture of oligomers with the same TAGs. These would of course be indistinguishable; we added a sentence to explain this prerequisite better in the manuscript (line 315-316):

It is important to note here that a mixture of oligomers with the same TAGs would of course be indistinguishable.

Comment: 6. You also never clearly state how many possibilities the Passerini reaction yields at each position in the tetramer. Presumably 3 possibilities at each position - but then how do you only arrive at 4 unique tetramer sequences when there should be 34 possible sequences?

Answer: As mentioned above, we used 12 possible sidechains (relating to 12 repeat units), that can freely be chosen for each position within the macromolecule. Thus, for a tetramer, 12^4 combinations are possible. The tetramers that we synthesized for each of the TAGs were selected in order to achieve the highest possible structural variety of sidechains/repeat units. We decided to synthesize four different tetramers for each TAG as a proof of concept as synthesizing all possible combinations (i.e. 20736 tetramers for each TAG) would be unachievable within a reasonable timeframe.

Comment: 7. You say that there are only four (presumably unique) tetramers. Why aren't you maximizing your information storage by varying the tetramer sequences?

Answer: Please also see comments above. Yes, we synthesized four different tetramers for each TAG as an example; however, we did vary the sequence in the tetramers in order to generate the highest possible chemical variety thus validating that it is very reasonable to assume that all possible tetramers can be prepared.

Comment: 8. I don't understand the basis for this sentence "The manual analysis of the MS/MS results was important as we needed to ensure that the herein used set of aldehydes did not produce overlapping mass fragments that would hinder the unambiguous assignment of all peaks." What does this have to do with analysis by-hand? Wouldn't an automated analysis yield the same result? Why can't you simply pre-check your aldehyde masses before synthesis so that the aldehydes and their adducts don't yield overlapping peaks?

Answer: Thank you for this comment. Of course, an automated and a "by-hand" analysis should lead to exactly the same result. It was thus part of our study to first investigate how the oligomers fragment and if a characteristic fragmentation pattern exists, or not. By manually reading out the sequences, we generated information about how the molecules fragment, which was crucial information for establishing a computer program, i.e. for programming the used algorithm (the program needed to be fed with the respective information about the breaking points within the molecule). In order to make reading easier, we have added a paragraph to the main text explain this in more detail (lines 184-192). Indeed, it is possible to check some potentially overlapping fragments by pre-calculation before the syntheses, which we did. However, one cannot always predict characteristic fragmentation patterns and since fragmentation can theoretically occur at many positions within the molecule, one cannot predict every fragment that is formed.

Comment: 9. Figure 2 seems to attempt to show the tetramer sequence variation, but confuses me even more. The key question is how many different units are possible at each monomer position? The figure seems to indicate through its colors that there are more than 3 possibilities at each position. This really needs clarification. Where does the number 12 side chains come from in your 12^6 permutations? It was never clearly enunciated what the 12 units are at each position. Towards this end, I find the sentence "twelve possible side chains

were taken into account, as their suitability for this synthesis approach was demonstrated with the tetramer library and the three hexamers, as well as the six repeating units...” If you have a hexamer, you have six repeating units, but if you have a tetramer, you have four. I don’t understand how tetramers and hexamer can demonstrate this.

Answer: Thank you for this comment. We are sorry that it was difficult to understand/read our manuscript and added a paragraph to the main text to clarify.

For our study, we use a well-established synthesis protocol developed by us, demonstrating that various different aldehydes can be used to introduce different side chains (und thus different repeating units) to the defined oligomers [(*Angew. Chem.*, **2016**, *128*, 1222-1225), (*Communication Chemistry.*, **2020**, *3*,63), (*Polym., Chem.* **2019**, *10*, 2716-2722)]. These components are freely selectable and can be introduced to any desired position in the sequence. Theoretically, it would be possible to introduce even more side chains as every aldehyde that does not cause side reactions in the Passerini reaction could potentially be used. However, we did not want to claim a theoretical number for the herein presented calculations and decided to use the number of aldehydes that we have demonstrated to be applicably for our synthesis route. Consequently, the number twelve corresponds to the number of aldehydes that have been used in exactly the same synthesis protocol before as well as herein and this number also represents the freely selectable repeating units at each position of the described sequence-defined oligomers.

With this synthesis protocol, various aldehydes can be used to introduce different sidechains.^[6,8,29] Thus, different repeating units can be introduced to the defined oligomers at predefined positions. This procedure is well established, and all experimental details are described in the supplementary information (sections **3.3**, **3.4**, **3.5**). Therefore, the already established isocyanide **M1** with a benzyl ester protected acid group was synthesized in a three step synthesis (see supplementary information **3.1**) .^[6] Using this approach, twelve different aldehydes **14a-l** (see supplementary information), to ensure side chain variation, were carefully selected to allow the simplified read-out of the sequence by tandem mass spectrometry by avoiding those aldehydes potentially yielding identical mass fragments. The aldehydes can be introduced to any desired position of the sequence, we were able to demonstrate the application of twelve different aldehydes in the synthesis of the oligomers. Consequently, the number of the aldehydes represents the freely selectable repeating units at each position of the sequence defined oligomers.

Minor Edits

Comment: 1. Line 113, page 4 has a reference error

Answer: Thank you for this remark. There was indeed a mistake which we corrected now.

Reviewer 3

The authors report on the synthesis of information-containing sequence-defined oligomers via the Passerini protocol, pioneered by this group already many years ago, in which a certain amount of data can be stored and read out by MS/MS. Whereas the synthesis and read-out

as such are of no advancements, the novelty claimed in this work is that by simply mixing a few of such uniform oligomers, the data storage capacity can be significantly increased, i.e. up to 65 bit. Hence, more information can be stored in shorter oligomers, thereby by-passing synthetic constraints associated to the preparation of long sequences. To enable a straightforward read-out of the discrete mixture of oligomers, specific 'mass markers' were incorporated at the start of each sequence so to allow for ESI-MS/MS characterization to be performed manually, which after successful completion could be translated into an automated read-out script.

Answer: We thank the reviewer for this accurate summary.

Comment: The main issue I have is the conceptual approach that is presented by the mixing of the sequences and the claimed increase in data storage capacity because of this. Taking into account a definition of data storage capacity as being 'the amount of data that can be stored in a storage device', I would argue that the so-called device here is the synthesised sequence-defined oligomer. Hence, simply mixing 3 tetramers, in my opinion, does not allow one to increase the data storage capacity of the macromolecules themselves. In fact, in terms of data capacity, I do not see any difference between the mixture or when one would analyse the three sequences separately. Would the stored data within one molecule not be the same in both cases? What might be claimed, however, is that the data storage density has seen an increase. Indeed, the amount of data will become more concentrated when the three solutions of oligomers are mixed in a same volume compared to the three separate solutions. Then again, a clear definition of what is interpreted as data storage density (e.g. the amount of data per mL) should be included within the current study.

Answer: We can follow this argument to some extent, yet we have to disagree on some points. We report herein a concept, we never claimed a device – we used the term data storage system, and the system is comprised of three sequence-defined oligomers, each bearing an individual mass TAG. Thus, by mixing these three oligomers, they can be unambiguously identified and their individual data storage capacity can be combined.

Of course, one could analyze the three oligomers separately, but in this case they would be physically separated, the analysis performed step-wise. In our opinion, the latter is not one unified system. The reviewer already agreed in their statement that the data storage density in solution (assuming the same concentration) would be higher in the case of mixtures, but why would the capacity of this solution not be higher as the capacity of three separate solutions. We disagree on this argument, but tried to clarify this point in the manuscript text.

Comment: Overall, the presented work is of high quality and the supporting information is technically sound and should enable to reproduce the findings. However, the claims are often massively overstated in certain places. It is therefore believed to be crucial for the authors to rework their manuscript in order to better place it with regard to the state-of-the-art in the field and also provide a more honest reflection of the advancements and claims that are made. Should this be done appropriately and the listed concerns (above and below) should have been addressed, I would highly recommend this study for publication in Communications Chemistry.

Answer: We thank the reviewer for this statement and re-worked the manuscript accordingly (please see comments above and below).

Comment: 1. The motivation that “the selection of halogenated TAGs provides the molecule a characteristic isotopic pattern, unique for each TAG, that allows to unambiguously assign the mass of an investigated oligomer to a certain TAG” raises an important concern. TAG3 containing a Cl is a clever choice to introduce a distinct isotopic mass pattern in HRMS [in fact, the Du Prez group has recently introduced multiple halogens in the context of molecular data storage to facilitate data read-out, yet a fair comparison/reference to this work - although cited by the authors in the introduction (ref.2) – is, albeit very relevant, missing during this discussion].

Answer: Thanks for this comment. We have added a section to the manuscript in order to compare our work with the work of the Du Prez group more thoroughly.

Interestingly, the group of Du Prez reported the use of halogenated TAGs to write a pin code recently.^[2] There, a mono chlorinated, a mono brominated and a di-brominated indole were used, in addition to a non-halogenated indole. By using ESI-MS and the specific isotopic pattern, it was possible to carry out the read-out without tandem mass analysis.^[2]

Comment: This aside, TAG1 and TAG2 should not be referred to as ‘a specifically designed mass marker’, given the absence of a unique fingerprint when using F since fluor-19 is the only abundant/stable isotope. Thus, frankly any (commercially available) starting carboxylic acid with the slightest difference in mass would be capable of providing a sufficient m/z difference. It definitely makes one question the unlogical choice for synthesising the perfluoro-alkyl acids. While indeed TAG2 can be distinguished from TAG3 based on the absence/presence of a Cl, no such differentiation is possible between TAG1 and TAG2 based on the isotopic pattern. Replacing one of these latter tags with a bromine-containing marker would significantly increase the importance of this conceptual work and should preferentially be demonstrated. In case this would not be feasible (e.g. in view of the current global situation), I believe the authors should at least address this remark within the manuscript.

Answer: This is a legitimate question. It is correct that the choice of a brominated carboxylic acid would introduce another characteristic isotopic pattern. However, for the proof of concept, the mass difference between the two perfluorinated TAGs 1 and 2 was sufficiently high (164 g/mol) to distinguish the two TAGs unambiguously. This would also be possible with other commercially available acids, like stearic acid for instance. The use of a perfluorinated TAG was however interesting to us, since the analysis *via* ¹⁹F-NMR offers another straightforward method for analysis and the purification of F-Tag labelled oligomers is simplified (*Nat Commun.*, **2018**, *9*, 1439). For a mixture of more than three oligomers, we fully agree that it would be highly beneficial to introduce TAGs with characteristic and different isotope patterns, like the mentioned brominated acid. We added a remark concerning this discussion to the manuscript.

For the proof of concept, the mass differences between the perfluorinated TAG1 and TAG2 were sufficient (164 g/mol) to distinguish the two TAGs. This would also be possible with other commercially available acids, like stearic acid, but the use of perfluorinated acids was preferred because of the simplified workup as demonstrated previously.^[36]

Comment: 2. The authors mention tandem mass spectrometry to be a promising tool to read out information contained in macromolecules. Many other techniques are also capable of extracting information, yet can be more accessible, cost-effective and render more straightforward data to interpret. It is recommended to provide the reader with at least a few other state-of-the-art techniques that have been used within the field and elaborate why tandem MS is considered most applicable in the context of this study.

Answer: Thank you for this remark. We agree that other techniques are discussed (*Angew. Chem. Int. Ed.*, **2014**, *53*, 13010 – 13019). However, mainly tandem MS was demonstrated and other options are thus more a theoretical possibility than a real-life option, at least for the moment. The group of Du Prez demonstrated a successful read-out *via* ESI-MS with the use of halogenated TAGs and their specific isotopic pattern (*Adv. Sci.*, **2020**, *7*, 1903698), as discussed above and now more explicitly mentioned in the manuscript. We have added a short paragraph about other methods for extracting information from data-containing macromolecules to the manuscript.

Also, other analytic methods are discussed for a possible read-out. For example, Du Prez *et al.* reported in 2020 the usage of ESI-MS for a successful read-out.^[2]

Comment: 3. Making oligomers longer to increase data storage capacity is claimed to have synthetic limitations because of the rapidly decreasing cumulative/overall yields that are obtained after each iterative growth cycle. This is of course correct and will be an important consideration to make, but is this anyhow problematic? Given the vast amounts of scalable protocols and the limited amount of product that is anyhow required in data storage applications, a few milligrams of a 20-mer should suffice. As such, I fail to understand why a linear synthesis of an 18-mer would be much more strenuous than making three times a linear synthesis effort to obtain three different 6-mers. It is believed important to clarify this within the manuscript.

Answer: We agree and disagree with this statement. Indeed, it would be possible to synthesize an 18-mer, but this would require a considerable synthetic effort in our case. We aim for higher amounts of substances, that can be fully and unambiguously characterized, which is very difficult with only a few milligrams. Furthermore, depending in the target application, sometimes a few mg might be sufficient, but for other applications more material might be needed (for instance if large batches of a product should be labelled with an anti-counterfeit tags, as discussed in the literature, grams to kilograms might be necessary).

For future applications, the synthesis amounts higher than required for MS/MS analysis might be needed and was thus demonstrated here.

Comment: 4. On multiple occasions, the authors highlight an ‘excellent’, ‘very high’, etc. purity of the synthesised oligomers. These terms are open to interpretation and should thus be more precisely defined (i.e. expressed in %). One could argue about the use of SEC to evaluate purity, but then again, the same holds for LCMS. Would a combination of both not be more appropriate to assess the presence of by-products?

Answer: Thank you for this remark. We provide the purity of the oligomers in % now. We recently demonstrated that, using SEC, it is possible to detect impurities in very small quantities starting from 2% (*Polym. J.*, **2020**, *52*, 165-178). The SEC system used in our study is equipped with columns specifically designed for low molecular weight oligomers. Furthermore, we could show in a previous project (*Polym. Chem.*, **2019**, *10*, 3859-3867) that with higher molecular weights the detection limit of LC-MS is reached at some point because of the increasing polarity of the oligomers, while SEC gave much more accurate results and the impurities could still be detected. Therefore, in our opinion, SEC is the suitable tool for proving purity.

More generally, we agree that we should not exaggerate and corrected the manuscript accordingly throughout.

The high purity and molecular integrity of the oligomers is confirmed by mass spectrometry, NMR as well as SEC, whereby SEC is able to detect impurities as low as 2% with our setup.^[47]

Comment: 5. Related to the previous concern, is there a certain requirement on the purity that needs to be reached in order to uphold the reading-out of the oligomer mixture? In other words, which levels of impurities can be tolerated during the tandem MS analysis? This is an important question to ask in order for one to justify the extensive purification efforts that have been undertaken in this study.

Answer: For this project, the proof of concept was important. Therefore, the highest possible purity of the molecules was necessary for a clear analysis of the read-out. However, other groups have already demonstrated that a certain level of purity can be tolerated during tandem MS analysis. We would assume that comparable amounts of impurities would also be tolerable for the analysis of our mixtures, as the fragmentation is only performed for distinct molecule peaks of the MS spectrum. Therefore, impurities do not have such an influence on the fragmentation. A more detailed analysis of this aspect would be interesting for future work.

Comment: 6. The absence of compound numbers in the manuscript (e.g. isocyanide M1, Passerini product 43) makes the discussion regarding the oligomer synthesis hard to follow. Please include compound numbers where seen fit.

Answer: Thank you for the remark. We have added an explanation to the manuscript in order to clarify.

Comment: 7. The authors claim ‘a tetramer’ can be synthesised within 8 working days. This claim seems a bit purposely formulated to be misleading. Looking at the Supporting Information, T1 requires at least 14 days to synthesise (not taking into account the time

needed for purification in between every growth cycle). While 8 days might be sufficient to make one certain tetramer in particular, it needs to be specified to what compound this applies exactly. Perhaps providing a time frame that covers all prepared oligomers allows for a more honest evaluation of the synthetic efforts?

Answer: Indeed, there was a mistake, thank you for pointing this out. For tetramer T9, 9 days are necessary for the synthesis of this specific tetramer. Reaction times are often confusing, because the reactions were stirred over the weekend when started on a Friday, although the reaction itself is completed after 6 – 8 hours for short sequences up to tetramers (*Polym. Chem.*, **2019**, *10*, 3859-3867). However, to allow for a candid evaluation of the synthesis effort, we changed this part in the manuscript (line 155).

Comment: 8. Upon growing the oligomers longer, much longer reaction times are needed to reach full conversion for the Passerini reaction. This is definitely a considerable drawback of this iterative growth protocol. Can these reaction times be shortened by optimising the reaction conditions (e.g. solvent, heating, catalyst)?

Answer: We are in general very experienced with the Passerini reaction and have optimized it extensively for different components, different solvents, different reaction times, on solid phase and in solution etc. In a previous study, the reaction was followed by online-IR (*Polym. Chem.*, **2019**, *10*, 3859-3867) showing that the reaction time increases significantly with increasing length of the oligomer. In our opinion, this is due to entanglement and we doubt that this can be much optimized. Besides, the reaction time was not the focus of this work.

Comment: 9. Although this does not make a difference in calculating the amount of bit that can be stored, should the max. number of possible permutations upon mixing the 3 separate hexamers not be $12^6 + 12^6 + 12^6 = 3 \cdot 12^6$, instead of $12^6 * 12^6 * 12^6 = 12^{18}$?

Answer: First of all, the number of permutations of course makes a very significant difference in the final amount of bits that can be stored, since the number of bits is calculated from the permutations with equation 2 (see manuscript). Indeed, the correct way to calculate it is: $12^6 \times 12^6 \times 12^6 = 12^{6+6+6} = 12^{18}$, i.e. the permutations are not additive. This results in a significantly higher number of permutations.

Comment: 10. Fig. 3c: an offset instead of an overlay would make the interpretation more clear.

Answer: Thank you for this comment. We have changed the figure accordingly.

Comment: 11. “The algorithm attempts to reconstruct a molecule composed out of these components, whose fragments can be found in the given mass spectrum, according to equation 2 “; eq. 2 allows for the calculation of the storage capacity and is not related to the read-out using the discussed algorithm. Please adapt.

Answer: Thank you for this remark. There was indeed a mistake, which we corrected now.

Comment: 12. A one-to-one comparison of the automated read-out programme is made with the one previously developed by Du Prez and co-workers. A (brief) comparison with regard to the programme by Lutz and co-workers (Macromolecules 2017, 50, 20, 8290–8296) might be a good addition to this.

Answer: Thank you for this comment. We added a brief comparison between our and the program of Lutz and coworkers (Macromolecules 2017, 50, 20, 8290–8296) to the manuscript.

Moreover, Lutz *et al.* described a “millisecond sequencing” of binary coded polymers using a program with implemented algorithm.^[38,48] This algorithm searches for the mass of the starter molecule. Afterwards, the mass of the starter plus the mass of the first backbone plus one of the possible side chains must be found. Subsequently, the next repeating unit is checked, and so on.^[48] For the binary system, this easy algorithm worked well. Compared to our program, due to the use of a variety of different side chains, it was necessary to develop another algorithm. A “filter system” and some criteria, for example fragments without the mass of the TAGs, had furthermore to be implemented to our algorithm to allow for the analysis of our more complex structures.

REVIEWERS' COMMENTS:

Reviewer #1 (Remarks to the Author):

The authors have made sufficient revisions and the response is satisfactory. It can be publishable in its current form.

Reviewer #2 (Remarks to the Author):

The authors have worked hard to address all of the reviewer comments and to clarify many of the points in their manuscript. In particular, I appreciated their efforts to more clearly state and illustrate how they generated their sequence diversity. As someone working from the information theory side of the coin, I believe the manuscript would still benefit from including a sentence detailing their explicit calculation of bits. As can be seen from the discussion among the reviewers, even chemists involved in information processing tend not to know all of the ins and outs of bit counting. Nevertheless, I now believe that the manuscript is ready for publication.

As an additional comment, I would encourage the authors to think further about how they can use the presence/absence of polymers in their mixtures to increase their capacity. Otherwise, as was stated by the other reviewers, their only, albeit possibly important, argument for using mixtures is to ease synthesis. Regardless of synthesis, their storage capacity would increase if they considered keeping or throwing out certain polymers.

Reviewer #3 (Remarks to the Author):

I have reviewed the resubmission and thank the authors for their answers to my comments. I believe the additional information that has been provided in their response can be important to any reader that would like to gain a more in-depth understanding of the presented conceptual system. I continue to express my recommendation for this study to be included in Communications Chemistry.